# Learning Disentangled Representations with Semi-Supervised Deep Generative Models

**N. Siddharth**[†]
University of Oxford
nsid@robots.ox.ac.uk

**Brooks Paige**[†]
Alan Turing Institute
University of Cambridge
bpaige@turing.ac.uk

**Jan-Willem van de Meent**[†]
Northeastern University
j.vandemeent@northeastern.edu

**Alban Desmaison**
University of Oxford
alban@robots.ox.ac.uk

**Noah D. Goodman**
Stanford University
ngoodman@stanford.edu

**Pushmeet Kohli** [*]
Deepmind
pushmeet@google.com

**Frank Wood**
University of Oxford
fwood@robots.ox.ac.uk

**Philip H.S. Torr**
University of Oxford
philip.torr@eng.ox.ac.uk

## Abstract

Variational autoencoders (VAEs) learn representations of data by jointly training a probabilistic encoder and decoder network. Typically these models encode all features of the data into a single variable. Here we are interested in learning disentangled representations that encode distinct aspects of the data into separate variables. We propose to learn such representations using model architectures that generalise from standard VAEs, employing a general graphical model structure in the encoder and decoder. This allows us to train partially-specified models that make relatively strong assumptions about a subset of interpretable variables and rely on the flexibility of neural networks to learn representations for the remaining variables. We further define a general objective for semi-supervised learning in this model class, which can be approximated using an importance sampling procedure. We evaluate our framework's ability to learn disentangled representations, both by qualitative exploration of its generative capacity, and quantitative evaluation of its discriminative ability on a variety of models and datasets.

## 1 Introduction

Learning representations from data is one of the fundamental challenges in machine learning and artificial intelligence. Characteristics of learned representations can depend on their intended use. For the purposes of solving a single task, the primary characteristic required is suitability for that task. However, learning separate representations for each and every such task involves a large amount of wasteful repetitive effort. A representation that has some factorisable structure, and consistent semantics associated to different parts, is more likely to generalise to a new task.

Probabilistic generative models provide a general framework for learning representations: a model is specified by a joint probability distribution both over the data and over latent random variables, and a representation can be found by considering the posterior on latent variables given specific data. The learned representation — that is, inferred values of latent variables — depends then not just on the data, but also on the generative model in its choice of latent variables and the relationships between the latent variables and the data. There are two extremes of approaches to constructing generative models. At one end are fully-specified probabilistic graphical models [19, 22], in which a practitioner decides on all latent variables present in the joint distribution, the relationships between them, and the functional form of the conditional distributions which define the model. At the other end are

---

[*]Author was at Microsoft Research during this project. † indicates equal contribution.

deep generative models [8, 17, 20, 21], which impose very few assumptions on the structure of the model, instead employing neural networks as flexible function approximators that can be used to train a conditional distribution on the data, rather than specify it by hand.

The tradeoffs are clear. In an explicitly constructed graphical model, the structure and form of the joint distribution ensures that latent variables will have particular semantics, yielding a *disentangled* representation. Unfortunately, defining a good probabilistic model is hard: in complex perceptual domains such as vision, extensive feature engineering (e.g. Berant et al. [1], Siddharth et al. [31]) may be necessary to define a suitable likelihood function. Deep generative models completely sidestep the difficulties of feature engineering. Although they address learning representations which then enable them to better reconstruct data, the representations themselves do not always exhibit consistent meaning along axes of variation: they produce *entangled* representations. While such approaches have considerable merit, particularly when faced with the absence of any side information about data, there are often situations when aspects of variation in data can be, or are desired to be characterised.

Bridging this gap is challenging. One way to enforce a disentangled representation is to hold different axes of variation fixed during training [21]. Johnson et al. [14] combine a neural net likelihood with a conjugate exponential family model for the latent variables. In this class of models, efficient marginalisation over the latent variables can be performed by learning a projection onto the same conjugate exponential family in the encoder. Here we propose a more general class of *partially-specified* graphical models: probabilistic graphical models in which the modeller only needs specify the exact relationship for some subset of the random variables in the model. Factors left undefined in the model definition are then learned, parametrised by flexible neural networks. This provides the ability to situate oneself at a particular point on a *spectrum*, by specifying precisely those axes of variations (and their dependencies) we have information about or would like to extract, and learning disentangled representations for them, while leaving the rest to be learned in an entangled manner.

A subclass of partially-specified models that is particularly common is that where we can obtain supervision data for some subset of the variables. In practice, there is often variation in the data which is (at least conceptually) easy to explain, and therefore annotate, whereas other variation is less clear. For example, consider the MNIST dataset of handwritten digits: the images vary both in terms of content (which digit is present), and style (how the digit is written), as is visible in the right-hand side of Fig. 1. Having an explicit "digit" latent variable captures a meaningful and consistent axis of variation, independent of style; using a partially-specified graphical model means we can define a "digit" variable even while leaving unspecified the semantics of the different styles, and the process of rendering a digit to an image. With unsupervised learning there is no guarantee that inference on a model with 10 classes will induce factored latent representations with factors corresponding to the the 10 digits. However, given a small amount of labelled examples, this task becomes significantly easier. Fundamentally, our approach conforms to the idea that well-defined notions of disentanglement require specification of a task under which to measure it [4]. For example, when considering images of people's faces, we might wish to capture the person's identity in one context, and the lighting conditions on the faces in another, facial features in another, or combinations of these in yet other contexts. Partially-specified models and weak supervision can be seen as a way to operationalise this task-dependence directly into the learning objective.

In this paper we introduce a recipe for learning and inference in partially-specified models, a flexible framework that learns disentangled representations of data by using graphical model structures to encode constraints to interpret the data. We present this framework in the context of *variational autoencoders* (VAEs), developing a generalised formulation of semi-supervised learning with DGMs that enables our framework to automatically employ the correct factorisation of the objective for any given choice of model and set of latents taken to be observed. In this respect our work extends previous efforts to introduce supervision into variational autoencoders [18, 24, 32]. We introduce a variational objective which is applicable to a more general class of models, allowing us to consider graphical-model structures with arbitrary dependencies between latents, continuous-domain latents, and those with dynamically changing dependencies. We provide a characterisation of how to compile partially-supervised generative models into stochastic computation graphs, suitable for end-to-end training. This approach allows us also *amortise* inference [7, 23, 29, 34], simultaneously learning a network that performs approximate inference over representations at the same time we learn the unknown factors of the model itself. We demonstrate the efficacy of our framework on a variety of tasks, involving classification, regression, and predictive synthesis, including its ability to encode latents of variable dimensionality.

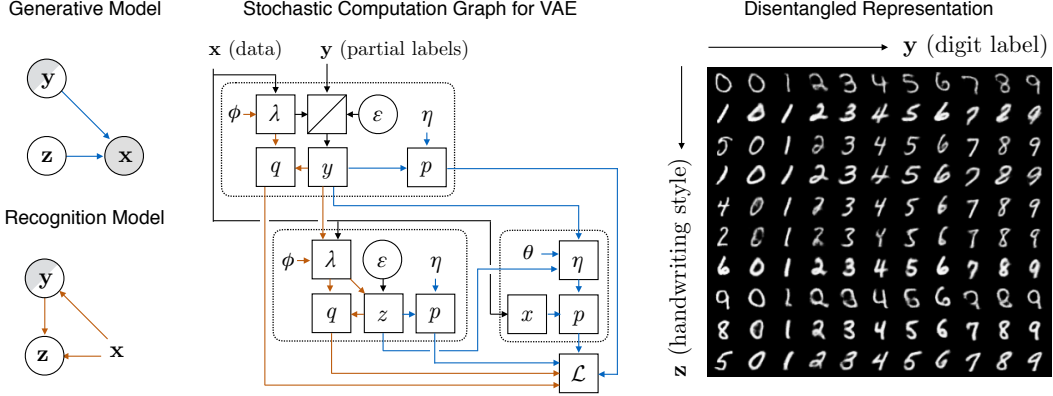

Figure 1: Semi-supervised learning in structured variational autoencoders, illustrated on MNIST digits. **Top-Left**: Generative model. **Bottom-Left:** Recognition model. **Middle**: Stochastic computation graph, showing expansion of each node to its corresponding sub-graph. Generative-model dependencies are shown in blue and recognition-model dependencies are shown in orange. See Section 2.2 for a detailed explanation. **Right:** learned representation.

## 2    Framework and Formulation

VAEs [17, 28] are a class of deep generative models that simultaneously train both a probabilistic encoder and decoder for a elements of a data set $\mathcal{D} = \{\mathbf{x}^1, \ldots \mathbf{x}^N\}$. The central analogy is that an encoding $\mathbf{z}$ can be considered a latent variable, casting the decoder as a conditional probability density $p_\theta(\mathbf{x}|\mathbf{z})$. The parameters $\eta_\theta(\mathbf{z})$ of this distribution are the output of a deterministic neural network with parameters $\theta$ (most commonly MLPs or CNNs) which takes $\mathbf{z}$ as input. By placing a weak prior over $\mathbf{z}$, the decoder defines a posterior and joint distribution $p_\theta(\mathbf{z} \mid \mathbf{x}) \propto p_\theta(\mathbf{x} \mid \mathbf{z})p(\mathbf{z})$.

Inference in VAEs can be performed using a variational method that approximates the posterior distribution $p_\theta(\mathbf{z} \mid \mathbf{x})$ using an encoder $q_\phi(\mathbf{z} \mid \mathbf{x})$, whose parameters $\lambda_\phi(\mathbf{x})$ are the output of a network (with parameters $\phi$) that is referred to as an "inference network" or a "recognition network". The generative and inference networks, denoted by solid and dashed lines respectively in the graphical model, are trained jointly by performing stochastic gradient ascent on the *evidence lower bound* (ELBO) $\mathcal{L}(\phi, \theta; \mathcal{D}) \leq \log p_\theta(\mathcal{D})$,

$$\mathcal{L}(\phi, \theta; \mathcal{D}) = \sum_{n=1}^{N} \mathcal{L}(\phi, \theta; \mathbf{x}^n) = \sum_{n=1}^{N} \mathbb{E}_{q_\phi(\mathbf{z}|\mathbf{x}^n)}[\log p_\theta(\mathbf{x}^n \mid \mathbf{z}) + \log p(\mathbf{z}) - \log q_\phi(\mathbf{z}|\mathbf{x}^n)]. \quad (1)$$

Typically, the first term $\mathbb{E}_{q_\phi(\mathbf{z}|\mathbf{x}^n)}[\log p_\theta(\mathbf{x}^n \mid \mathbf{z})]$ is approximated by a Monte Carlo estimate and the remaining two terms are expressed as a divergence $-\mathrm{KL}(q_\phi(\mathbf{z}|\mathbf{x}^n)\|p(\mathbf{z}))$, which can be computed analytically when the encoder model and prior are Gaussian.

In this paper, we will consider models in which both the generative model $p_\theta(\mathbf{x}, \mathbf{y}, \mathbf{z})$ and the approximate posterior $q_\phi(\mathbf{y}, \mathbf{z} \mid \mathbf{x})$ can have arbitrary conditional dependency structures involving random variables defined over a number of different distribution types. We are interested in defining VAE architectures in which a subset of variables $\mathbf{y}$ are interpretable. For these variables, we assume that supervision labels are available for some fraction of the data. The VAE will additionally retain some set of variables $\mathbf{z}$ for which inference is performed in a fully unsupervised manner. This is in keeping with our central goal of defining and learning in partially-specified models. In the running example for MNIST, $\mathbf{y}$ corresponds to the classification label, whereas $\mathbf{z}$ captures all other implicit features, such as the pen type and handwriting style.

This class of models is more general than the models in the work by Kingma et al. [18], who consider three model designs with a specific conditional dependence structure. We also do not require $p(\mathbf{y}, \mathbf{z})$ to be a conjugate exponential family model, as in the work by Johnson et al. [15]. To perform semi-supervised learning in this class of models, we need to i) define an objective that is suitable to general dependency graphs, and ii) define a method for constructing a stochastic computation graph [30] that incorporates both the conditional dependence structure in the generative model and that of the recognition model into this objective.

## 2.1 Objective Function

Previous work on semi-supervised learning for deep generative models [18] defines an objective over $N$ unsupervised data points $D = \{\mathbf{x}^1, \ldots, \mathbf{x}^N\}$ and $M$ supervised data points $\mathcal{D}^{\mathrm{sup}} = \{(\mathbf{x}^1, \mathbf{y}^1), \ldots, (\mathbf{x}^M, \mathbf{y}^M)\}$,

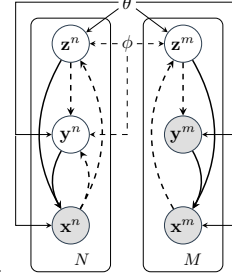

$$\mathcal{L}(\theta, \phi; \mathcal{D}, \mathcal{D}^{\mathrm{sup}}) = \sum_{n=1}^{N} \mathcal{L}(\theta, \phi; \mathbf{x}^n) + \gamma \sum_{m=1}^{M} \mathcal{L}^{\mathrm{sup}}(\theta, \phi; \mathbf{x}^m, \mathbf{y}^m). \quad (2)$$

Our model's joint distribution factorises into unsupervised and supervised collections of terms over $\mathcal{D}$ and $\mathcal{D}^{\mathrm{sup}}$ as shown in the graphical model. The standard variational bound on the joint evidence of all observed data (including supervision) also factorises as shown in Eq. (2). As the factor corresponding to the unsupervised part of the graphical model is exactly that as Eq. (1), we focus on the supervised term in Eq. (2), expanded below, incorporating an additional weighted component as in Kingma et al. [18].

$$\mathcal{L}^{\mathrm{sup}}(\theta, \phi; \mathbf{x}^m, \mathbf{y}^m) = \mathbb{E}_{q_\phi(\mathbf{z}|\mathbf{x}^m, \mathbf{y}^m)}\left[\log \frac{p_\theta(\mathbf{x}^m, \mathbf{y}^m, \mathbf{z})}{q_\phi(\mathbf{z} \mid \mathbf{x}^m, \mathbf{y}^m)}\right] + \alpha \log q_\phi(\mathbf{y}^m \mid \mathbf{x}^m). \quad (3)$$

Note that the formulation in Eq. (2) introduces an constant $\gamma$ that controls the relative strength of the supervised term. While the joint distribution in our model implicitly weights the two terms, in situations where the relative sizes of $\mathcal{D}$ and $\mathcal{D}^{\mathrm{sup}}$ are vastly different, having control over the relative weights of the terms can help ameliorate such discrepancies.

This definition in Eq. (3) implicitly assumes that we can evaluate the conditional probability $q_\phi(\mathbf{z}|\mathbf{x}, \mathbf{y})$ and the marginal $q_\phi(\mathbf{y}|\mathbf{x}) = \int d\mathbf{z}\, q_\phi(\mathbf{y}, \mathbf{z}|\mathbf{x})$. This was indeed the case for the models considered by Kingma et al. [18], which have a factorisation $q_\phi(\mathbf{y}, \mathbf{z}|\mathbf{x}) = q_\phi(\mathbf{z}|\mathbf{x}, \mathbf{y})q_\phi(\mathbf{y}|\mathbf{x})$.

Here we will derive an estimator for $\mathcal{L}^{\mathrm{sup}}$ that generalises to models in which $q_\phi(\mathbf{y}, \mathbf{z} \mid \mathbf{x})$ can have an arbitrary conditional dependence structure. For purposes of exposition, we will for the moment consider the case where $q_\phi(\mathbf{y}, \mathbf{z} \mid \mathbf{x}) = q_\phi(\mathbf{y} \mid \mathbf{x}, \mathbf{z})q_\phi(\mathbf{z} \mid \mathbf{x})$. For this factorisation, generating samples $\mathbf{z}^{m,s} \sim q_\phi(\mathbf{z} \mid \mathbf{x}^m, \mathbf{y}^m)$ requires inference, which means we can no longer compute a simple Monte Carlo estimator by sampling from the unconditioned distribution $q_\phi(\mathbf{z} \mid \mathbf{x}^m)$. Moreover, we also cannot evaluate the density $q_\phi(\mathbf{z} \mid \mathbf{x}^m, \mathbf{y}^m)$.

In order to address these difficulties, we re-express the supervised terms in the objective as

$$\mathcal{L}^{\mathrm{sup}}(\theta, \phi; \mathbf{x}^m, \mathbf{y}^m) = \mathbb{E}_{q_\phi(\mathbf{z}|\mathbf{x}^m, \mathbf{y}^m)}\left[\log \frac{p(\mathbf{x}^m, \mathbf{y}^m, \mathbf{z})}{q_\phi(\mathbf{y}^m, \mathbf{z} \mid \mathbf{x}^m)}\right] + (1 + \alpha) \log q_\phi(\mathbf{y}^m \mid \mathbf{x}^m), \quad (4)$$

which removes the need to evaluate $q_\phi(\mathbf{z} \mid \mathbf{x}^m, \mathbf{y}^m)$. We can then use (self-normalised) importance sampling to approximate the expectation. To do so, we sample proposals $\mathbf{z}^{m,s} \sim q_\phi(\mathbf{z} \mid \mathbf{x}^m)$ from the unconditioned encoder distribution, and define the estimator

$$\mathbb{E}_{q_\phi(\mathbf{z}|\mathbf{x}^m, \mathbf{y}^m)}\left[\log \frac{p_\theta(\mathbf{x}^m, \mathbf{y}^m, \mathbf{z})}{q_\phi(\mathbf{y}^m, \mathbf{z} \mid \mathbf{x}^m)}\right] \simeq \frac{1}{S}\sum_{s=1}^{S} \frac{w^{m,s}}{Z^m} \log \frac{p_\theta(\mathbf{x}^m, \mathbf{y}^m, \mathbf{z}^{m,s})}{q_\phi(\mathbf{y}^m, \mathbf{z}^{m,s} \mid \mathbf{x}^m)}, \quad (5)$$

where the unnormalised importance weights $w^{m,s}$ and normaliser $Z^m$ are defined as

$$w^{m,s} := \frac{q_\phi(\mathbf{y}^m, \mathbf{z}^{m,s} \mid \mathbf{x}^m)}{q_\phi(\mathbf{z}^{m,s} \mid \mathbf{x}^m)}, \qquad\qquad Z^m = \frac{1}{S}\sum_{s=1}^{S} w^{m,s}. \quad (6)$$

To approximate $\log q_\phi(\mathbf{y}^m \mid \mathbf{x}^m)$, we use a Monte Carlo estimator of the lower bound that is normally used in maximum likelihood estimation,

$$\log q_\phi(\mathbf{y}^m \mid \mathbf{x}^m) \geq \mathbb{E}_{q_\phi(\mathbf{z}|\mathbf{x}^m)}\left[\log \frac{q_\phi(\mathbf{y}^m, \mathbf{z} \mid \mathbf{x}^m)}{q_\phi(\mathbf{z} \mid \mathbf{x}^m)}\right] \simeq \frac{1}{S}\sum_{s=1}^{S} \log w^{m,s}, \quad (7)$$

using the same samples $\mathbf{z}^{m,s}$ and weights $w^{m,s}$ as in Eq. (5). When we combine the terms in Eqs. (5) and (7), we obtain the estimator

$$\hat{\mathcal{L}}^{\mathrm{sup}}(\theta, \phi; \mathbf{x}^m, \mathbf{y}^m) := \frac{1}{S}\sum_{s=1}^{S} \frac{w^{m,s}}{Z^m} \log \frac{p_\theta(\mathbf{x}^m, \mathbf{y}^m, \mathbf{z}^{m,s})}{q_\phi(\mathbf{y}^m, \mathbf{z}^{m,s} \mid \mathbf{x}^m)} + (1 + \alpha) \log w^{m,s}. \quad (8)$$

We note that this estimator applies to *any* conditional dependence structure. Suppose that we were to define an encoder $q_\phi(\mathbf{z}_2, \mathbf{y}_1, \mathbf{z}_1 \mid \mathbf{x})$ with factorisation $q_\phi(\mathbf{z}_2 \mid \mathbf{y}_1, \mathbf{z}_1, \mathbf{x}) q_\phi(\mathbf{y}_1 \mid \mathbf{z}_1, \mathbf{x}) q_\phi(\mathbf{z}_1 \mid \mathbf{x})$. If we propose $\mathbf{z}_2 \sim q_\phi(\mathbf{z}_2 \mid \mathbf{y}_1, \mathbf{z}_1, \mathbf{x})$ and $\mathbf{z}_1 \sim q_\phi(\mathbf{z}_1 \mid \mathbf{x})$, then the importance weights $w^{m,s}$ for the estimator in Eq. (8) are defined as

$$w^{m,s} := \frac{q_\phi(\mathbf{z}_2^{m,s}, \mathbf{y}_1^m, \mathbf{z}_1^{m,s} \mid \mathbf{x}^m)}{q_\phi(\mathbf{z}_2^{m,s} \mid \mathbf{y}_1^m, \mathbf{z}_1^{m,s}, \mathbf{x}^m) q_\phi(\mathbf{z}_1^{m,s} \mid \mathbf{x}^m)} = q_\phi(\mathbf{y}_1^m \mid \mathbf{z}_1^{m,s}, \mathbf{x}^m).$$

In general, the importance weights are simply the product of conditional probabilities of the supervised variables $\mathbf{y}$ in the model. Note that this also applies to the models in Kingma et al. [18], whose objective we can recover by taking the weights to be constants $w^{m,s} = q_\phi(\mathbf{y}^m \mid \mathbf{x}^m)$.

We can also define an objective analogous to the one used in importance-weighted autoencoders [2], in which we compute the logarithm of a Monte Carlo estimate, rather than the Monte Carlo estimate of a logarithm. This objective takes the form

$$\hat{\mathcal{L}}^{\text{sup,iw}}(\theta, \phi; \mathbf{x}^m, \mathbf{y}^m) := \log \left[ \frac{1}{S} \sum_{s=1}^S \frac{p_\theta(\mathbf{x}^m, \mathbf{y}^m, \mathbf{z}^{m,s})}{q_\phi(\mathbf{z}^{m,s} \mid \mathbf{x}^m)} \right] + \alpha \log \left[ \frac{1}{S} \sum_{s=1}^S w^{m,s} \right], \qquad (9)$$

which can be derived by moving the sums in Eq. (8) into the logarithms and applying the substitution $w^{m,s} / q_\phi(\mathbf{y}^m, \mathbf{z}^{m,s} \mid \mathbf{x}^m) = 1 / q_\phi(\mathbf{z}^{m,s} \mid \mathbf{x}^m)$.

## 2.2 Construction of the Stochastic Computation Graph

To perform gradient ascent on the objective in Eq. (8), we map the graphical models for $p_\theta(\mathbf{x}, \mathbf{y}, \mathbf{z})$ and $q_\phi(\mathbf{y}, \mathbf{z} | \mathbf{x})$ onto a stochastic computation graph in which each stochastic node forms a sub-graph. Figure 1 shows this expansion for the simple VAE for MNIST digits from [17]. In this model, $\mathbf{y}$ is a discrete variable that represents the underlying digit, our latent variable of interest, for which we have partial supervision data. An unobserved Gaussian-distributed variable $\mathbf{z}$ captures the remainder of the latent information. This includes features such as the hand-writing style and stroke thickness. In the generative model (Fig. 1 top-left), we assume a factorisation $p_\theta(\mathbf{x}, \mathbf{y}, \mathbf{z}) = p_\theta(\mathbf{x} \mid \mathbf{y}, \mathbf{z}) p(\mathbf{y}) p(\mathbf{z})$ in which $\mathbf{y}$ and $\mathbf{z}$ are independent under the prior. In the recognition model (Fig. 1 bottom-left), we use a conditional dependency structure $q_\phi(\mathbf{y}, \mathbf{z} \mid \mathbf{x}) = q_{\phi_\mathbf{z}}(\mathbf{z} \mid \mathbf{y}, \mathbf{x}) q_{\phi_\mathbf{y}}(\mathbf{y} | \mathbf{x})$ to disentangle the digit label $\mathbf{y}$ from the handwriting style $\mathbf{z}$ (Fig. 1 right).

The generative and recognition model are jointly form a stochastic computation graph (Fig. 1 centre) containing a sub-graph for each stochastic variable. These can correspond to fully supervised, partially supervised and unsupervised variables. This example graph contains three types of sub-graphs, corresponding to the three possibilities for supervision and gradient estimation:

- For the fully supervised variable $\mathbf{x}$, we compute the likelihood $p$ under the generative model, that is $p_\theta(\mathbf{x} \mid \mathbf{y}, \mathbf{z}) = \mathcal{N}(\mathbf{x}; \eta_\theta(\mathbf{y}, \mathbf{z}))$. Here $\eta_\theta(\mathbf{y}, \mathbf{z})$ is a neural net with parameters $\theta$ that returns the parameters of a normal distribution (i.e. a mean vector and a diagonal covariance).
- For the unobserved variable $\mathbf{z}$, we compute both the prior probability $p(\mathbf{z}) = \mathcal{N}(\mathbf{z}; \eta_\mathbf{z})$, and the conditional probability $q_\phi(\mathbf{z} \mid \mathbf{x}, \mathbf{y}) = \mathcal{N}(\mathbf{z}; \lambda_{\phi_\mathbf{z}}(\mathbf{x}, \mathbf{y}))$. Here the usual reparametrisation is used to sample $\mathbf{z}$ from $q_\phi(\mathbf{z} \mid \mathbf{x}, \mathbf{y})$ by first sampling $\epsilon \sim \mathcal{N}(\mathbf{0}, \mathbf{I})$ using the usual reparametrisation trick $\mathbf{z} = g(\epsilon, \lambda_\phi(\mathbf{x}, \mathbf{y}))$.
- For the partially observed variable $\mathbf{y}$, we also compute probabilities $p(\mathbf{y}) = \text{Discrete}(\mathbf{y}; \eta_\mathbf{y})$ and $q_{\phi_\mathbf{y}}(\mathbf{y} | \mathbf{x}) = \text{Discrete}(\mathbf{y}; \lambda_{\phi_\mathbf{z}}(\mathbf{x}))$. The value $\mathbf{y}$ is treated as observed when available, and sampled otherwise. In this particular example, we sample $\mathbf{y}$ from a $q_{\phi_\mathbf{y}}(\mathbf{y} | \mathbf{x})$ using a Gumbel-softmax [13, 25] relaxation of the discrete distribution.

The example in Fig. 1 illustrates a general framework for defining VAEs with arbitrary dependency structures. We begin by defining a node for each random variable. For each node we then specify a distribution type and parameter function $\eta$, which determines how the probability under the generative model depends on the other variables in the network. This function can be a constant, fully deterministic, or a neural network whose parameters are learned from the data. For each unsupervised and semi-supervised variable we must additionally specify a function $\lambda$ that returns the parameter values in the recognition model, along with a (reparametrised) sampling procedure.

Given this specification of a computation graph, we can now compute the importance sampling estimate in Eq. (8) by simply running the network forward repeatedly to obtain samples from $q_\phi(\cdot | \lambda)$ for all unobserved variables. We then calculate $p_\theta(\mathbf{x}, \mathbf{y}, \mathbf{z})$, $q_\phi(\mathbf{y} | \mathbf{x})$, $q_\phi(\mathbf{y}, \mathbf{z} | \mathbf{x})$, and the importance

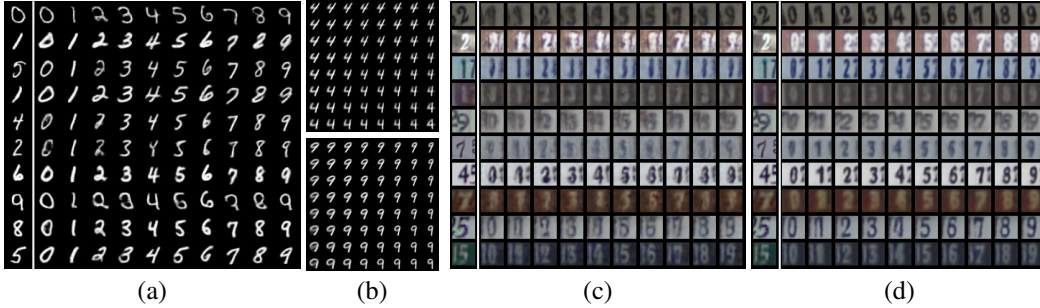

Figure 2: (a) Visual analogies for the MNIST data, partially supervised with just 100 labels (out of 50000). We infer the style variable **z** and then vary the label **y**. (b) Exploration in style space with label **y** held fixed and (2D) style **z** varied. Visual analogies for the SVHN data when (c) partially supervised with just 1000 labels, and (d) fully supervised.

weight $w$, which is the joint probability of all semi-supervised variable for which labels are available. This estimate can then be optimised with respect to the variables $\theta$ and $\phi$ to train the autoencoder.

## 3   Experiments

We evaluate our framework along a number of different axes pertaining to its ability to learn disentangled representations through the provision of partial graphical-model structures for the latents and weak supervision. In particular, we evaluate its ability to (i) function as a classifier/regressor for particular latents under the given dataset, (ii) learn the generative model in a manner that preserves the semantics of the latents with respect to the data generated, and (iii) perform these tasks, in a flexible manner, for a variety of different models and data.

For all the experiments run, we choose architecture and parameters that are considered standard for the type and size of the respective datasets. Where images are concerned (with the exception of MNIST), we employ (de)convolutional architectures, and employ a standard GRU recurrence in the Multi-MNIST case. For learning, we used AdaM [16] with a learning rate and momentum-correction terms set to their default values. As for the mini batch sizes, they varied from 100-700 depending on the dataset being used and the sizes of the labelled subset $\mathcal{D}^{\text{sup}}$. All of the above, including further details of precise parameter values and the source code, including our PyTorch-based library for specifying arbitrary graphical models in the VAE framework, is available at –
https://github.com/probtorch/probtorch.

### 3.1   MNIST and SVHN

We begin with an experiment involving a simple dependency structure, in fact the very same as that in Kingma et al. [18], to validate the performance of our importance-sampled objective in the special case where the recognition network and generative models factorise as indicated in Fig. 1(left), giving us importance weights that are constant $w^{m,s} = q_\phi(y^m|x^m)$. The model is tested on it's ability to classify digits and perform conditional generation on the MNIST and Google Street-View House Numbers (SVHN) datasets. As Fig. 1(left) shows, the generative and recognition models have the "digit" label, denoted **y**, partially specified (and partially supervised) and the "style" factor, denoted **z**, assumed to be an unobserved (and unsupervised) variable.

Figure 2(a) and (c) illustrate the conditional generation capabilities of the learned model, where we show the effect of first transforming a given input (leftmost column) into the disentangled latent space, and with the style latent variable fixed, manipulating the digit through the generative model to generate data with expected visual characteristics. Note that both these results were obtained with partial supervision – 100 (out of 50000) labelled data points in the case of MNIST and 1000 (out of 70000) labelled data points in the case of SVHN. The style latent variable **z** was taken to be a diagonal-covariance Gaussian of 10 and 15 dimensions respectively. Figure 2(d) shows the same for SVHN with full supervision. Figure 2(b) illustrates the alternate mode of conditional generation, where the style latent, here taken to be a 2D Gaussian, is varied with the digit held fixed.

Next, we evaluate our model's ability to effectively learn a classifier from partial supervision. We compute the classification error on the label-prediction task on both datasets, and the results are

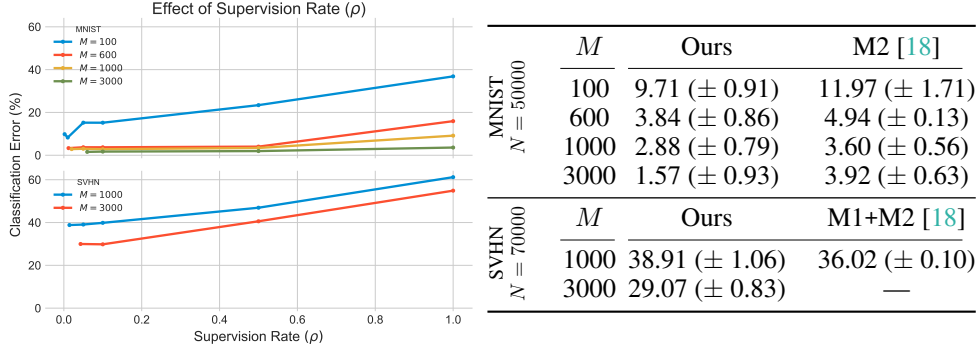

Figure 3: **Right:** Classification error rates for different labelled-set sizes $M$ over multiple runs, with supervision rate $\rho = \frac{\gamma M}{N + \gamma M}, \gamma = 1$. For SVHN, we compare against a multi-stage process (M1+M2) [18], where our model only uses a single stage. **Left:** Classification error over different labelled set sizes and supervision rates for MNIST (top) and SVHN (bottom). Here, scaling of the classification objective is held fixed at $\alpha = 50$ (MNIST) and $\alpha = 70$ (SVHN). Note that for sparsely labelled data ($M \ll N$), a modicum of over-representation ($\gamma > 1$) helps improve generalisation with better performance on the test set. Conversely, too much over-representation leads to overfitting.

reported in the table in Fig. 3. Note that there are a few minor points of difference in the setup between our method and those we compare against [18]. We always run our models directly on the data, with no pre-processing or pre-learning on the data. Thus, for MNIST, we compare against model M2 from the baseline which does just the same. However, for SVHN, the baseline method does not report errors for the M2 model; only the two-stage M1+M2 model which involves a separate feature-extraction step on the data before learning a semi-supervised classifier.

As the results indicate, our model and objective does indeed perform on par with the setup considered in Kingma et al. [18], serving as basic validation of our framework. We note however, that from the perspective of achieving the lowest possible classification error, one could adopt any number of alternate factorisations [24] and innovations in neural-network architectures [27, 33].

**Supervision rate:** As discussed in Section 2.1, we formulate our objective to provide a handle on the relative weight between the supervised and unsupervised terms. For a given unsupervised set size $N$, supervised set size $M$, and scaling term $\gamma$, the relative weight is $\rho = \gamma M/(N + \gamma M)$. Figure 3 shows exploration of this relative weight parameter over the MNIST and SVHN datasets and over different supervised set sizes $M$. Each line in the graph measures the classification error for a given $M$, over $\rho$, starting at $\gamma = 1$, i.e. $\rho = M/(N + M)$. In line with Kingma et al.[18], we use $\alpha = 0.1/\rho$. When the labelled data is very sparse ($M \ll N$), over-representing the labelled examples during training can help aid generalisation by improving performance on the test data. In our experiments, for the most part, choosing this factor to be $\rho = M/(N + M)$ provides good results. However, as is to be expected, over-fitting occurs when $\rho$ is increased beyond a certain point.

## 3.2 Intrinsic Faces

We next move to a more complex domain involving generative models of faces. As can be seen in the graphical models for this experiment in Fig. 4, the dependency structures employed here are more complex in comparison to those from the previous experiment. Here, we use the "Yale B" dataset [6] as processed by Jampani et al. [12] for the results in Fig. 5. We are interested in showing that our model can learn disentangled representations of identity and lighting and evaluate it's performance on the tasks of (i) classification of person identity, and (ii) regression for lighting direction.

Note that our generative model assumes no special structure – we simply specify a model where all latent variables are independent under the prior. Previous work [12] assumed a generative model with latent variables identity $i$, lighting $l$, shading $s$, and reflectance $r$, following the relationship $(n \cdot l) \times r + \epsilon$ for the pixel data. Here, we wish to demonstrate that our generative model still learns the correct relationship over these latent variables, by virtue of the structure in the recognition model and given (partial) supervision.

Note that in the recognition model (Fig. 4), the lighting $l$ is a latent variable with *continuous* domain, and one that we partially supervise. Further, we encode identity $i$ as a categorical random variable,

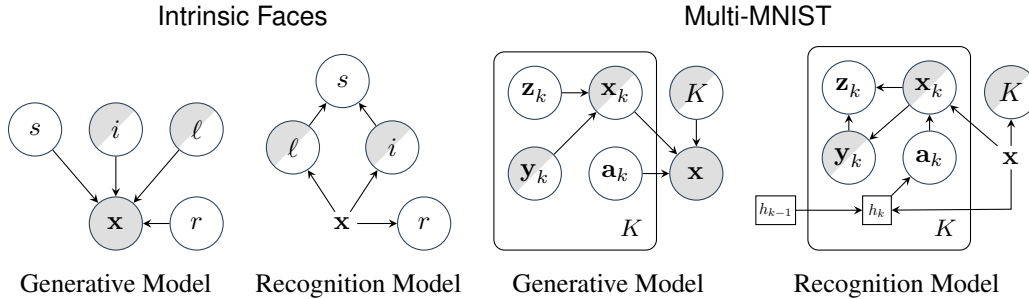

Figure 4: Generative and recognition models for the intrinsic-faces and multi-MNIST experiments.

| | | Identity | Lighting |
|---|---|---|---|
| Ours (Full Supervision) | | 1.9% ($\pm$ 1.5) | 3.1% ($\pm$ 3.8) |
| Ours (Semi-Supervised) | | 3.5% ($\pm$ 3.4) | 17.6% ($\pm$ 1.8) |
| Jampani et al. [12] (plot asymptotes) | | $\approx 30$ | $\approx 10$ |

Figure 5: **Left:** Exploring the generative capacity of the supervised model by manipulating identity and lighting given a fixed (inferred) value of the other latent variables. **Right:** Classification and regression error rates for identity and lighting latent variables, fully-supervised, and semi-supervised (with 6 labelled example images for each of the 38 individuals, a supervision rate of $\rho = 0.5$, and $\alpha = 10$). Classification is a direct 1-out-of-38 choice, whereas for the comparison, error is a nearest-neighbour loss based on the inferred reflectance. Regression loss is angular distance.

instead of constructing a pixel-wise surface-normal map (each assumed to be independent Gaussian) as is customary. This formulation allows us to address the task of predicting identity directly, instead of applying surrogate evaluation methods (e.g. nearest-neighbour classification based on inferred reflectance). Figure 5 presents both qualitative and quantitative evaluation of the framework to jointly learn both the structured recognition model, and the generative model parameters.

### 3.3 Multi-MNIST

Finally, we conduct an experiment that extends the complexity from the prior models even further. Particularly, we explore the capacity of our framework to handle models with *stochastic* dimensionality – having the number of latent variables itself determined by a random variable, and models that can be composed of other smaller (sub-)models. We conduct this experiment in the domain of multi-MNIST. This is an apposite choice as it satisfies both the requirements above – each image can have a varying number of individual digits, which essentially dictates that the model must learn to count, and as each image is itself composed of (scaled and translated) exemplars from the MNIST data, we can employ the MNIST model itself within the multi-MNIST model.

The model structure that we assume for the generative and recognition networks is shown in Fig. 4. We extend the models from the MNIST experiment by composing it with a stochastic sequence generator, in which the loop length $K$ is a random variable. For each loop iteration $k = 1, \ldots, K$, the generative model iteratively samples a digit $\mathbf{y}_k$, style $\mathbf{z}_k$, and uses these to generate a digit image $\mathbf{x}_k$ in the same manner as in the earlier MNIST example. Additionally, an affine tranformation is also sampled for each digit in each iteration to transform the digit images $\mathbf{x}_k$ into a common, combined canvas that represents the final generated image $\mathbf{x}$, using a spatial transformer network [11].

In the recognition model, we predict the number of digits $K$ from the pixels in the image. For each loop iteration $k = 1, \ldots, K$, we define a Bernoulli-distributed digit image $\mathbf{x}_k$. When supervision is available, we compute the probability of $\mathbf{x}_k$ from the binary cross-entropy in the same manner as in the likelihood term for the MNIST model. When no supervision is available, we deterministically set $\mathbf{x}_k$ to the mean of the distribution. This can be seen akin to providing bounding-boxes around the constituent digits as supervision, which must be taken into account when learning the affine transformations that decompose a multi-MNIST image into its constituent MNIST-like images. This model design is similar to the one used in DRAW [10], recurrent VAEs [3], and AIR [5].

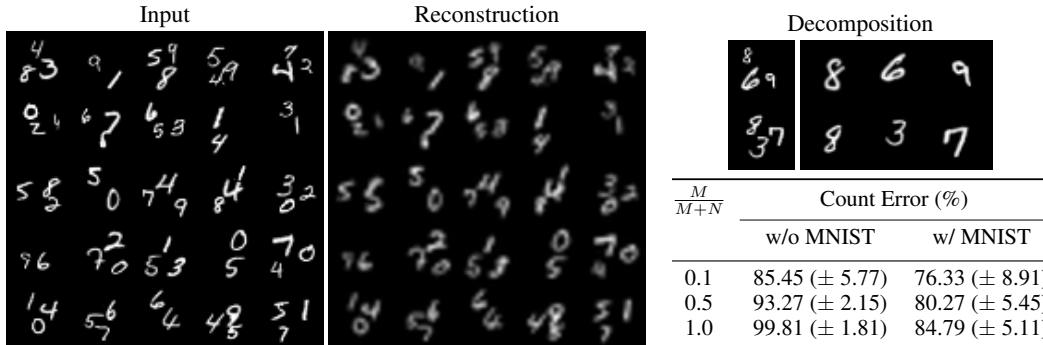

| | Input | Reconstruction | | Decomposition | |

| $\frac{M}{M+N}$ | Count Error (%) | |
|---|---|---|
| | w/o MNIST | w/ MNIST |
| 0.1 | 85.45 ($\pm$ 5.77) | 76.33 ($\pm$ 8.91) |
| 0.5 | 93.27 ($\pm$ 2.15) | 80.27 ($\pm$ 5.45) |
| 1.0 | 99.81 ($\pm$ 1.81) | 84.79 ($\pm$ 5.11) |

Figure 6: **Left:** Example input multi-MNIST images and reconstructions. **Top-Right**: Decomposition of Multi-MNIST images into constituent MNIST digits. **Bottom-Right:** Count accuracy over different supervised set sizes $M$ for given dataset size $M + N = 82000$.

In the absence of a canonical multi-MNIST dataset, we created our own from the MNIST dataset by manipulating the scale and positioning of the standard digits into a combined canvas, evenly balanced across the counts (1-3) and digits. We then conducted two experiments within this domain. In the first experiment, we seek to measure how well the stochastic sequence generator learns to count on its own, with no heed paid to disentangling the latent representations for the underlying digits. Here, the generative model presumes the availability of individual MNIST-digit images, generating combinations under sampled affine transformations. In the second experiment, we extend the above model to now also incorporate the same *pre-trained* MNIST model from the previous section, which allows the generative model to sample MNIST-digit images, while also being able to predict the underlying digits. This also demonstrates how we can leverage compositionality of models: when a complex model has a known simpler model as a substructure, the simpler model and its learned weights can be dropped in directly.

The count accuracy errors across different supervised set sizes, reconstructions for a random set of inputs, and the decomposition of a given set of inputs into their constituent individual digits, are shown in Fig. 6. All reconstructions and image decompositions shown correspond to the nested-model configuration. We observe that not only are we able to reliably infer the counts of the digits in the given images, we are able to simultaneously reconstruct the inputs as well as its constituent parts.

## 4 Discussion and Conclusion

In this paper we introduce a framework for learning disentangled representations of data using partially-specified graphical model structures and semi-supervised learning schemes in the domain of variational autoencoders (VAEs). This is accomplished by defining hybrid generative models which incorporate both structured graphical models and unstructured random variables in the same latent space. We demonstrate the flexibility of this approach by applying it to a variety of different tasks in the visual domain, and evaluate its efficacy at learning disentangled representations in a semi-supervised manner, showing strong performance. Such partially-specified models yield recognition networks that make predictions in an interpretable and disentangled space, constrained by the structure provided by the graphical model and the weak supervision.

The framework is implemented as a PyTorch library [26], enabling the construction of stochastic computation graphs which encode the requisite structure and computation. This provides another direction to explore in the future — the extension of the stochastic computation graph framework to probabilistic programming [9, 35, 36]. Probabilistic programs go beyond the presented framework to permit more expressive models, incorporating recursive structures and higher-order functions. The combination of such frameworks with neural networks has recently been studied in Le et al. [23] and Ritchie et al. [29], indicating a promising avenue for further exploration.

### Acknowledgements

This work was supported by the EPSRC, ERC grant ERC-2012-AdG 321162-HELIOS, EPSRC grant Seebibyte EP/M013774/1, and EPSRC/MURI grant EP/N019474/1. BP & FW were supported by The Alan Turing Institute under the EPSRC grant EP/N510129/1. JWM, FW & NDG were supported under DARPA PPAML through the U.S. AFRL under Cooperative Agreement FA8750-14-2-0006. JWM was additionally supported through startup funds provided by Northeastern University. FW was additionally supported by Intel and DARPA D3M, under Cooperative Agreement FA8750-17-2-0093.

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
