[Reviews · NeurIPS 2017]

Reviewer 1



This work extends [17], the original semi-supervised VAE proposed by Kingma et al. Instead of consider the label y’ as a partially observed variable, they also disentangle y from z, which is part of the latent variables. This yields a loss function that is different from the original paper. Since this variable is discrete, the author uses Gumbel-Softmax to optimize it. To my understanding this loss is more interesting since every unlabled sample also contributes to the learning of p(y|x) instead of only the supervised ones. On the other side, the experiments are very week * MNIST: The original results of Kingma are better and there are lot’s of even stronger results in the litterature * SVHN: There is a single point of comparison where the new approach seems to be significantly better than [17] * Intrinsic Faces: A single point of comparison with the original paper [11] without mentioning any other work on this dataset. * Multi-MNIST: The results are interesting but can’t be compared to other approaches since the dataset is specific to this work. Overall, the idea is good, but author ignore any other work that has been done on semi-supervised VAE in the 3 years following [17]. And the experimental results are rather week and seems to be indicating that the new approach does not perform better the the original one.

Reviewer 2



The authors develop a framework allowing VAE type computation on a broad class of probablistic model structures. This is motivated in particular by the idea that some lvs may have a straightforward meaning and have some labels available (e.g. which digit in MNIST/SVHN, what lighting direction in the face data), whereas others are more intangible (e.g. writing style in MNIST). They propose a slightly different approach to the semi-supervised VAE of Kingma et al., by considering the (semi)supervised variables y as LVs forced to specific values for the supervised data samples. This is straightforward in the setting where q(y|x) can be calculated directly, and can be handled by importance sampling if integration over z is required to calculate q(y|x). Experiments are presented on MNIST, SVHN and a faces image data with variation in lighting according 38 individuals. On all three dataset the authors show successful qualitative learning of the semisupervised LVs. Quantitatively the performance on MNIST doesn't quite match that of Kingma et al. While I'm not too worried about this it would be interesting to know the authors' guess as to the cause - a difference in architecture? the objective itself? On SVHN however they do somewhat better. On the faces dataset it's not clear that the baseline is more than a strawman, but presumably other methods haven't been evaluated here? The paper is very clearly written for the most part. There are a few places where more explanation/detail would have been beneficial (the page limit makes this challenging), in particular - I was unclear about the faces dataset - is the lighting effectively an angle in [0,360]? What is "n" in the Equation used by the original publication? Possibly a surface tangent? - The description of the RNN used for counting is very terse. Minor comments - Saliman & Knowles 2012 demonstrated representing q as a hierarchy of LVs so should probably be cited. - "Composing graphical models with neural networks for structured representations and fast inference" appears twice in the citations. - I don't know if I like "latents" instead of "lvs" but maybe I could get used to it.

Reviewer 3



This paper investigates the use of models mixing ideas from 'classical' graphical models (directed graphical models with nodes with known meaning, and with known dependency structures) and deep generative models (learned inference network, learned factors implemented as neural networks). A particular focus is the use of semi-supervised data and structure to induce 'disentangling' of features. They test on a variety of datasets and show that disentangled features are interpretable and can be used for downstream tasks. In contrast with recent work on disentangling ("On the Emergence of Invariance and Disentangling in Deep Representations", "Early visual concept learning with unsupervised deep learning", and others), here the disentangling is not really an emergent property of the model's inductive bias, and more related to the (semi)-supervision provided on semantic variables of the model. My opinion on the paper is overall positive - the derivations are correct, the model and ideas contribute to the intersection of graphical models / probabilistic modeling and deep learning, and well-made/detailed experiments support the authors' claims. Negative: - I don't think the intersection of ideas in structured graphical models and deep generative model is particularly novel at this point (see many of the paper's own references). In particular I am not convinced by the claims of increased generality. Clearly there is a variety of deep generative models that fall in the PGM-DL spectrum described by the authors and performing inference with flexible set of observed variables mostly involves using different inference networks (with potential parameter reuse for efficiency). Regarding continuous domain latents: I don't see what would prevent the semi-supervised model from Kingma et al from handling continous latent variable: wouldn't simply involve changing q(y|x) to a continuous distribution and reparametrize (just as they do for the never-supervised z)? Generally, methodologically the papers appears somewhat incremental (including the automated construction of the stochastic computation graph, which is similar to what something like Edward (Tran et al.) does). - Some numerical experiments results are slightly disappointing (the semi-supervised MNIST numbers seem worse than Kingma et al.; the multi-MNIST reconstructions are somewhat blurry; what do samples look like?) Positive: - The paper is well written and tells a compelling story. - The auxiliary bound introduced in (4)-(5) is novel and and allows the posterior in the semi-supervised case to force learning over the classification network (in corresponding equation (6) of Kingma et al. the classification network does not appear). Do the authors know how the variance between (4) and the analogous expression from Kingma et al compare? - The rest of the experiments (figure 2, 5) paint a clear picture of why having semantics inform the structure of the generative model can lead to interesting way to probe the behavior of our generative models.